# Regulation Mechanism and Potential Value of Active Substances in Spices in Alcohol–Liver–Intestine Axis Health

**DOI:** 10.3390/ijms25073728

**Published:** 2024-03-27

**Authors:** Jianyu Huang, Tao Huang, Jinjun Li

**Affiliations:** 1State Key Laboratory for Managing Biotic and Chemical Threats to the Quality and Safety of Agro-Products, Institute of Food Science, Zhejiang Academy of Agricultural Sciences, Hangzhou 310021, China; huangjianyunbu@163.com; 2College of Food and Pharmaceutical Science, Ningbo University, Ningbo 315211, China

**Keywords:** spice, active substance, hepatointestinal axis, alcohol

## Abstract

Excessive alcohol intake will aggravate the health risk between the liver and intestine and affect the multi-directional information exchange of metabolites between host cells and microbial communities. Because of the side effects of clinical drugs, people tend to explore the intervention value of natural drugs on diseases. As a flavor substance, spices have been proven to have medicinal value, but they are still rare in treating hepatointestinal diseases caused by alcohol. This paper summarized the metabolic transformation of alcohol in the liver and intestine and summarized the potential value of various perfume active substances in improving liver and intestine diseases caused by alcohol. It is also found that bioactive substances in spices can exert antioxidant activity in the liver and intestine environment and reduce the oxidative stress caused by diseases. These substances can interfere with fatty acid synthesis, promote sugar and lipid metabolism, and reduce liver injury caused by steatosis. They can effectively regulate the balance of intestinal flora, promote the production of SCFAs, and restore the intestinal microenvironment.

## 1. Introduction

The liver and gut undergo constant and complex role changes during human health and disease, and there is a bidirectional relationship between the intestine and its microbiota and the liver in response to dietary, genetic, and environmental signals [1,2,3]. This relationship is exhibited by material transport, with the portal vein acting as the primary anatomical structure for the communication between the liver and the intestine; dietary components, symbionts, and their metabolites circulate between them through the portal vein blood. Simultaneously, bile secreted by the liver is absorbed by the intestine and enters the portal vein blood, contributing to the bidirectional circulation of the liver and intestine [1,2,3]; this relationship is also apparent in immunomodulation, during which intestinal substances interact with hepatocytes, other hepatic parenchymal cells, and hepatic immune cells through blood flow in hepatic sinuses [4]. In spite of this, the liver can recruit and activate immune cells in response to intestinal metabolism or pathogen-derived signals [5]. Besides, bile acid (BA) is very important in regulating hepatointestinal homeostasis and promoting glucose and lipid metabolism, and completes circulation after passing through the liver, gallbladder, small intestine, colon, and plasma [6]. Intestinal flora, as an important part of enterohepatic circulation, regulates homeostasis through complex interaction with the host immune metabolism system [7]. Interestingly, the synergistic combination of BAs and intestinal flora is often an important basis for the clinical treatment of hepatointestinal diseases [6,8,9]. Additionally, the intestinal barrier can affect metabolism and immune regulation in the hepatointestinal axis [9,10,11]. The hepatotestinal axis is a dynamic system, and any part of it may cause disorder in the system.

It is reported that diet is the leading cause of hepatointestinal diseases, and excessive drinking has gradually become a chronic killer of human health. Alcohol and its metabolites will destroy the normal enterohepatic circulation function of the human body, increasing oxidative stress and lipid peroxide production, even causing intestinal microecology imbalance and increasing intestinal mucosal permeability, intestinal endotoxemia, and inflammatory factors. Typical diseases are alcoholic liver disease (ALD) and nonalcoholic liver disease (NAFLD) [12]. Drinking has become an expected behavior during social occasions and to relieve emotions. Excessive drinking can amplify the degree of free radicals and dangers to human fitness in the physique. Globally, an estimated 741,300 new cancer cases in 2020 were caused by drinking [13]. At the same time, the liver is the main organ of alcohol metabolism, and alcohol, as a xenobiotic, can be converted into acetaldehyde in the process of liver metabolism. This highly reactive substance can covalently bind with protein in the liver to change the functional structure, thus hindering alcohol metabolism [14], mainly manifested in decreased biological activities of antioxidant enzymes and signal transduction substances and the necrosis of liver cells. At the same time, excessive free radicals can cause lipid peroxidation and a decrease in mitochondrial respiration, thus aggravating liver damage [15]. Interestingly, alcohol can also affect the diversity of intestinal flora and even induce the production of a large amount of lipopolysaccharide (LPS), which binds with toll-like receptor 4 (TLR4) and enters the liver. While activating Kupffer cells, it also produces a variety of inflammatory factors. Liver injury [16] results in liver steatosis, steatohepatitis, liver fibrosis, cirrhosis, and even liver cancer [17,18]. At present, the main clinical treatment strategies are divided into anti-inflammatory cytokines and antioxidants, activating the immune system and changing intestinal flora. Among them, widely accepted schemes include abstinence from alcohol, corticosteroids, biological agents (such as anti-TNF-α drugs and accelerating the elimination of alcohol in the blood), and liver transplantation. Nevertheless, their functions are not comprehensive and even have many side effects on human health [19].

At present, there is no scientific remedy for hepatointestinal diseases caused by alcohol, and abstinence from alcohol is one of the primary means to alleviate the damage of alcohol to the liver and intestine [20]. In recent years, with the rapid development of natural products research, many plant extracts have entered the consumer market and played an increasingly important role in the treatment of alcoholic hepatointestinal diseases [21]. For example, bioactive substances such as polyphenols and polysaccharides in plants have been shown to improve liver damage, reduce the content of inflammatory factors, promote glucose and lipid metabolism, significantly regulate intestinal flora, and enhance intestinal mucosa [22]. Spices are a kind of food flavor substances that widely come from the roots, leaves, and fruits of plants. Nevertheless, many studies show that the active substances in spices have high medicinal value [23]. Recently, active ingredients in spices have been used in interventions for hepatointestinal diseases (Figure 1). However, few researchers have comprehensively described the active ingredients of spices and alcoholic hepatointestinal diseases. At the same time, the specific molecular mechanism needs to be well explained. Based on the above research gaps, this paper focuses on alcohol metabolism in the liver and intestine, including bile acids, hepatointestinal receptors, and intestinal flora. It summarizes the potential value of active substances in spices in improving hepatointestinal diseases. It provides a theoretical basis for further application in food and drug processing and clinical treatment.

## 2. Metabolic Mode of Alcohol in Liver

Exogenous alcohol smoothly enters the gastrointestinal tract after passing through the oral esophagus, is digested and absorbed into the blood, and most of the alcohol is metabolized in the liver. Of course, liver damage is directly related to the amount and frequency of alcohol use. In the investigation of Seulggie Choi and others, it is found that compared with light drinkers (male: 0.1 to 19.9 g/day; female: 0.1 to 9.9 g/day), heavy drinkers (male: ≥40.0 g/day; female: ≥20.0 g/day) have an increased risk of metabolic syndrome [24]. ALD is a typical example of the liver and intestinal diseases caused by alcohol. ALD initially shows asymptomatic steatosis, which is related to the formation of lipid droplets in the liver. Once more than 5% of liver cells contain lipid droplets, the disorder is classified as alcoholic fatty liver or steatosis [25,26]. ALD is a constellation of many diseases, from initial steatosis to steatohepatitis, liver fibrosis, and cirrhosis, and finally, liver cancer. It involves different pathological processes, mainly because alcohol can change the amount of fat produced in liver cells and induce intestinal secretion of endotoxins, thus promoting steatosis, oxidative stress, and liver cell damage [27].

Alcohol metabolism is closely associated with liver diseases. In general, the poisonous outcomes of alcohol on the body are related to the metabolic site. The gastrointestinal tract quickly absorbs alcohol entering the body. Approximately 10% of ingested alcohol is eliminated through sweat, breathing, and urine, and the liver oxidizes the rest; therefore, the liver is the hub of alcohol and its metabolites [28]. Alcohol abuse is reported to be directly related to fat metabolism in the liver, which may be because of the metabolic pathways of alcohol, namely oxidative metabolism and non-oxidative metabolism (Figure 2). That is, by inhibiting the activities of alcohol oxidase through components such as alcohol dehydrogenase (ADH), cytochrome P450, and catalase under aerobic conditions, alcohol metabolism changes from aerobic to anaerobic metabolism, thus increasing the content of fatty acid ethyl esters (FAEEs) in the body. In this chapter, the metabolic mechanism of alcohol in the liver is expounded from the perspective of biomolecules.

### 2.1. Oxidative Metabolic Pathway

When a small amount of alcohol is exposed to the surface of hepatocytes, hepatocytes secrete ADH and oxidize alcohol to form acetaldehyde, which causes the loss of hydrogen through the action of acetaldehyde dehydrogenase (ALDH) and coenzyme nicotinamide adenine dinucleotide (NAD+) to form acetate and nicotinamide adenine dinucleotide (NADH). Then, acetic acid is converted into acetyl coenzyme A, which is oxidized in the triphosphate cycle [29]. ADH and ALDH need NAD+ to transfer oxygen and reduce it to NADH [30]. Then, NADH and NAD are rapidly oxidized through the respiratory pathway to minimize the toxic effects of ethanol and related substances. Cytochrome P4502E1 (CYP2E1) and catalase (CAT) are also involved in alcohol metabolism [31]. Studies have shown that the peroxisome proliferator-activated receptor α (PPARα) can completely transform the pathway of reactive oxygen species (ROS) production induced by CYP2E1 into the pathway of ROS elimination induced by catalase during alcohol metabolism, thus accelerating alcohol elimination [32]. When hepatocytes are continuously exposed to alcohol, ethanol is converted into acetaldehyde through an enzymatic pathway, which reduces the ratio of NAD^+^/NADH and produces many ROS to destroy the body’s antioxidant function [33], thereby inducing the formation of lipid free radicals, destroying polyunsaturated fatty acids on hepatocyte membranes, causing lipid peroxidation, destroying cell membrane fluidity, and inducing mitochondrial damage to aggravate lipid accumulation in hepatocytes [34].

### 2.2. Non-Oxidative Metabolic Pathway

Abuse of alcohol increases the workload of the non-specific enzyme system, which results in the production of a large number of cytotoxic substances and causes liver and intestine dysfunction. Non-oxidative alcohol metabolism utilizes the combination of alcohol and endogenous metabolites such as glucuronic acid, sulfate, phospholipid, and fatty acid to produce ethyl glucuronide (EtG), ethyl sulfate (EtS), phosphatidylethanol (PEth), and fatty acid ethyl ester (FAEE) [35]. The retention time of non-oxidative alcohol metabolites in physique fluids and tissues is much longer than that of ethanol. Hence, they are often used as biomarkers for evaluating, screening, and diagnosing patients with excessive drinking in forensic medicine or the clinic [35,36]. EtG and EtS are stable and water-soluble. They are produced by the combination of ethanol and glucuronic acid through the action of uridine diphosphate glucuronosyltransferase (UGTs) and ethanol and sulfate through the action of cytoplasmic sulfotransferase (SULTs), singly [35,36]. Studies have shown that non-oxidative alcohol metabolites can also interfere with cell signal transduction and membrane-associated protein dysfunction [37,38].

## 3. The Influence of Alcohol Metabolism in Enterohepatic Circulation

The liver and intestine are the main battlefields of alcohol metabolism and absorption. The two-way communication between the intestine and the liver promotes the exchange of various substances, including alcohol metabolites, which is the direct reason alcohol affects the hepatointestinal axis. Of course, this process will change the nature and content of endogenous substances. For example, alcohol metabolism may lead to abnormal changes in the content and composition of BAs in the human body. In addition, drinking alcohol will change the composition and function of intestinal flora, thus affecting the production of short-chain fatty acids and the integrity of the intestinal barrier, and have a subtle impact on health [39,40]. However, at the molecular level, the crosstalk of alcohol on the hepatointestinal axis is still unclear. Therefore, this chapter will summarize the substance transformation, signal transduction, and intestinal flora changes of alcohol in the hepatointestinal axis (Figure 3).

### 3.1. Association between Alcohol and BAs

BAs are a group of amphipathic metabolites decomposed by cholesterol. It is an essential active component in bile and a critical nutritional signal transduction hormone derived from cholesterol [6,8,9,20]. It plays a vital role in maintaining the homeostasis of the body’s internal environment [22], promoting glucose and lipid metabolism [23], mobilizing immune function [41], affecting energy metabolism [42], and mediating inflammatory response [43]. Bile acid is an essential medium in enterohepatic circulation, and its transport is very subtle. Firstly, primary BAs were actively transported from hepatocytes to tubule space under the synergistic effect of the bile salt export pump (BSEP), multidrug resistance protein 3 (MDR3), and multidrug resistance protein 2 (MRP2) [44]. After reaching the intestine, the primary BAs produce metabolites (such as secondary BAs) under intestinal flora’s action to change the BA pool’s composition, make it more diversified and hydrophobic, and thus regulate the sugar and lipid metabolism of the host [45]. Finally, with the help of sodium taurocholate cotransporter polypeptide (NTCP, also known as SLC10A1) and organic anion transporter polypeptide 1 (OATP1), BAs were transported back to the liver through the superior mesenteric vein for subsequent circulation, and the unabsorbed BAs entered the colon. They were excreted in feces or dehydrogenated by intestinal bacteria [46]. This process is called enterohepatic circulation. As shown in Figure 4, BAs exist as signal molecules for the whole process, promoting homeostasis, substance metabolism, and transportation in the liver by activating nuclear and membrane G protein-coupled receptors [47]. The intestinal nuclear receptor ligand is mainly secondary BA reabsorbed by the intestine. It can interact with farnesol X receptor (FXR) [48], vitamin D receptor (VDR) [49], and pregnane X receptor (PXR) [50]. Through interactions between the heterodimer and retinoid X receptor (RXR), transcription factors are further activated, resulting in upregulation and downregulation of transcription by binding hormone response element (HRE) to promote the expression of related genes [51].

At present, the influence of alcohol on bile acids is very complicated, and many studies are mainly elaborated from these aspects. First, alcohol will promote the expression of the BA synthesis gene. This process is mainly due to the oxidative stress induced by alcohol in the liver, which activates cyclic adenylate response element binding protein H (CREBH) and changes the expression of many genes involved in BA metabolism [52]. For example, alcohol can promote CB1R signal transduction in the liver by up-regulating the endogenous cannabinoid 2-AG and inducing BA enzyme gene expression, thus leading to alcoholic steatohepatitis [53]. Secondly, drinking alcohol can disrupt the balance of bile acids by inducing the imbalance of intestinal flora involved in bile acid metabolism. Studies have proved that alcohol exposure can reduce the levels of taurine-bound Bas (TDCA, TCA, and TLCA) in the liver, duodenum, and ileum of rats and increase the levels of non-bound BAs and glycine-bound Bas (GCDCA, GDCA, GHDCA, and GLCA) with more substantial toxicity, which proves that the metabolism of BAs is related to alcoholic liver injury [54]. Muthiah et al. showed that the contents of taurocholate chenodeoxycholic acid (TCDCA) and taurocholate (TUDCA) in plasma were directly related to the severity of ALD. On the contrary, ursodeoxycholic acid in feces is negatively correlated with the severity of ALD [55]. Thirdly, alcohol can affect the expression of BA transporter and metabolism. In Guo et al.’s research, it was found that ethanol treatment can increase the mRNA level of BA efflux transporters (BSEP, MRP3/4, OSTα/β) in rats. On the contrary, the expression of NATC, another BA transporter, was inhibited [56]. In addition, liver cancer caused by drinking is closely related to the metabolic pathway of bile acids. For example, the research in Wenbo Chen et al. shows that alcohol not only activates oncogenes but reduces the expression of tumor suppressor genes. The bile salt export pump (BSEP) significantly reduced the activity of BAS transport. Nevertheless, the bile acid transporter gene also changed significantly [57]. Fourthly, drinking alcohol causes bile acid balance disorder by regulating bile acid receptor FXR. For example, research by Mingxing Huang and others showed that FXR knockout mice were more sensitive to ethanol-induced liver steatosis and inflammation. At the same time, ethanol treatment changed lipid metabolism, BAs homeostasis, and the expression of ethanol degradation genes in the liver [58].

Bile acid has been proven effective in regulating glucose, lipid metabolism, and immune homeostasis. Long-term drinking almost disrupts all aspects of liver lipid metabolism, such as inhibiting fatty acid oxidation and increasing abnormal lipid accumulation and triglyceride synthesis [59]. Studies have shown that alcohol can induce abnormal changes in genes involved in bile acid metabolism and even regulate FXR and TGR5 signal transmission, thus causing bile acid balance disorder, that is, changing the size of the bile acid pool in the liver and increasing the contents of the triglycerides and cholesterol [60]. Additionally, insulin resistance is a metabolic reaction of the body’s carbohydrates, lipids, and proteins [61]. There is a correlation between IR and BAS. For example, the research of Haeusler and others shows that the composition of the peripheral blood BA pool will change during IR, leading to the increase in 12a-hydroxylated BA (CA, DCA, and their combined forms) [62]. This increase can induce cholesterol absorption and aggravate dyslipidemia, diabetes, and obesity [63]. Similarly, in the experiment of Tiangang Li et al., it was found that the overexpression of CYP7A1 can promote the catabolism of liver cholesterol and the level of bile acid pool, increase the secretion of low-density lipoprotein, maintain the plasma triglyceride homeostasis, and even reduce the expression level of liver messenger RNA of several essential lipogenic and gluconeogenic genes, thus preventing IR to some extent [64].

### 3.2. Interactions between Alcohol and Intestinal Flora

There is a close relationship between intestinal microbial homeostasis and host health [65]. It is an important participant in enterohepatic circulation, and can affect the liver through different mechanisms, such as the production of metabolites (short-chain fatty acids), immunomodulation (immune response of the liver to intestinal-derived factors, such as lipopolysaccharide), bile acid metabolism (production of secondary bile acids) and changes in barrier integrity (tight junction protein). Previous studies have found that alcohol has changed the diversity of intestinal microbial communities, mainly by increasing the number of Bacteroides and reducing the ratio of Firmicutes to Bacteroides (F/B value) [66]. This result is also manifested in the human body. Bode et al. confirmed through culture experiments that alcohol leads to aerobic and anaerobic bacteria growth in the jejunum. Moreover, the aspirate of patients with alcoholism was found to contain high levels of gram-negative anaerobic bacteria and endospore rods [67]. Mutlu et al. found that the colonic microflora of alcoholics was disordered, and the median abundance of Bacteroides was low. In contrast, the median abundance of Proteus was high. Furthermore, changes in the colonic microflora in some alcoholics were persistent [68].

Alcohol can trigger insulin resistance through the type and metabolism of intestinal flora. Researchers have found that LPS and its cell surface receptor, toll-like receptor 4, (TLR4) can damage the sensitivity of alcohol-fed mice to insulin and increase resistance [69]. At the same time, Tiangang Li et al. found that insulin resistance can be reduced, and insulin receptor and receptor substrate expression can be increased by regulating intestinal flora and inhibiting LPS/TLR4/TNF-α signal transduction in the liver [70]. Importantly, Tadashi Takeuchi and others found that individuals with a high content of Chaetomium in the human intestine often have higher insulin resistance and higher fecal monosaccharide content; however, individuals with more Bacteroides have lower insulin resistance and lower monosaccharide content in feces [71]. Nevertheless, Yin-Yi Ding and others have proved through experiments that insulin resistance can be prevented by reducing the content of Firmicutes and increasing the content of Bacteroides and wart microflora [72]. These results indicate that increasing the number of intestinal-associated flora and reducing proinflammatory signal transduction can effectively intervene in insulin resistance. At the same time, alcohol intake can inhibit the related metabolic expression of bactericidal Leprosy C regenerated islet-derived protein 3b (Reg3b) and Reg3g in the small intestine, thus increasing the risk of ALD [73]. Moreover, long-term exposure to ethanol can increase the permeability of the intestinal mucosa and change the intestinal flora, thus intensifying the production of proinflammatory endotoxins [74], followed by migration of intestinal LPS to the liver and binding with LPS protein (LBP), differentiation cluster-14 (CD14), and myeloid differentiation factor-2. Activated KCs produce many ROS, inflammatory cytokines, and chemokines. An increased launch of proinflammatory cytokines and infiltration of other inflammatory cells eventually leads to liver damage [75].

The influence of alcohol on intestinal fungi is inevitable. Studies have shown that alcohol administration can increase the fungal community and the transfer of fungal products (β-glucan) to systemic circulation in mice. At the same time, β-glucan induces liver inflammation through CLEC7A, a lectin-like receptor on Kupffer cells and other possible bone marrow-derived cells. Subsequently, the increase in IL-1β expression and secretion leads to hepatocyte injury and promotes the development of ethanol-induced liver disease [76]. It is understood that the essential symbiotic fungal species in the human intestine are Candida, Saccharomyces cerevisiae, and Malassezia. Under the long-term symbiotic mode, the body is tolerant to fungi. However, when the intestinal barrier is broken, the human intestine becomes the primary target for fungi and their products [77]. An-Ming Yang et al. showed that Ethanol administration led to the overgrowth of intestinal fungi and increase in β-glucan plasma stages in mice and that the wide variety of fungi in the feces of mice increased significantly. Ethanol consumption increased the richness and diversity of fungal species, and the proportion of Pythium, Fusarium, and Aspergillus increased significantly; in contrast, the proportion of Candida decreased. Fungi and their metabolites can additionally cause liver cell damage [76]. Nevertheless, intestinal bacteria and fungal pathogens interact through related molecular patterns (PAMPs) to influence alcohol damage to the liver and intestine [78]. It can be proved that dynamic changes in the intestinal flora can reflect the severity of hepatointestinal sickness. For example, the relative abundance of Akkermansia decreases, and Veillonella increases in the intestines of patients with alcoholic hepatitis, which can indicate more severe illness [79].

### 3.3. Alcohol and the Influence of the Intestinal Mucosa

There is an essential intermediate boundary in the hepatointestinal axis, that is, the intestinal mucosal barrier. It is a congenital barrier that maintains intestinal homeostasis and isolates pathogenic bacteria and toxins [80]. It mainly consists of four parts, namely, mechanical, chemical, microbial, and immune barriers. The mechanical barrier is a selective permeability barrier composed of tight junctions (atresia protein, tight junction protein, and atresia zona 1 [ZO-1]), adhesive junctions, and desmosomes. The most significant effect is blocking the entry of bacteria, viruses, and endotoxins [81]. The chemical barrier consists of mucus secreted with intestinal epithelial cells, digestive juice, and antibacterial materials secreted via everyday bacteria [82]. The microbial barrier comprises the intestinal flora, mainly the mucosal and intestinal flora. The mucosal flora is composed primarily of Bifidobacterium and lactic acid bacteria. In contrast, the intestinal flora is mostly Escherichia coli and Enterococcus, which adhere to the intestinal mucosa and form multiple layers [83]. The immune barrier comprises intestinal-related lymphoid tissues, including intraepithelial lymphocytes, lamina propria lymphocytes, and intestinal lymph nodes. It contains the most immunoglobulin-secreting cells in the whole body and responds to the stimulation of the intestinal mucosa by antigens derived from bacteria, viruses, and toxins in the environment [65].

Alcohol abuse may destroy the characteristics of the intestinal mucosal barrier, including intestinal barrier tissue and mucosal-associated invariant T (MAIT), threatening the body’s immune system. Parlesak and others found that the concentration of endotoxin in the plasma of alcoholics improved by using greater than five times (*p* < 0.01) relative to that of non-alcoholics. Alcohol has been proven to break down the intestinal mucosal barrier and cause endotoxins to overflow [84]. At the same time, MAIT cell dysfunction can also contribute to innate and adaptive immune dysfunction induced by alcohol, which may lead to pathological changes in end organs [85]. Previous studies have shown that alcohol-related endotoxemia is mainly due to intestinal barrier dysfunction, which displaces LPS content from the intestinal cavity to systemic circulation [86]. Long-term drinking can lead to the thickening of the intestinal mucosa and, to an extent, in the expression of mucins such as mucin-2-2 (MUC2), which can aggravate damage to the intestinal barrier. Alcohol intake is also related to a decrease in Notch signaling and degrees of Notch ligands Dll1 and Dll4 in the ileum and colon and, to an extent, in the expression of Notch series genes such as Math1 and Spdef, which are specific to goblet cells [87]. It further interferes with intestinal antigen homeostasis in other ways. In response to alcohol intake, dendritic cells on the intestinal mucosa may be transported to mesenteric lymph nodes (MLNs) through mesenteric lymphatic vessels (MLVs), which promotes the proliferation of adipose tissue regulatory T cells (Treg) and promotes intestinal antigen homeostasis. Similarly, the change in intestinal permeability will further put the whole body in a state of low inflammation through metabolic endotoxemia, further aggravating the deterioration of glucose homeostasis and insulin resistance [88]. This process is due to the specific binding of LPS with TLR4, which triggers the activation of NF-κB and the release of proinflammatory factors in cells and activates serum kinases (JNK and IKK), which further induces serine phosphorylation of the insulin receptor substrate and intensifies insulin resistance [89].

### 3.4. Relationship between Alcohol and Receptor Function in the Liver and Intestine

Alcohol is the fuse to induce hepatointestinal diseases, and transmission of signals between the hepatointestinal axes and toxic metabolites produced by various cells are the media that trigger the body to open a line of defense, which further aggravates metabolic disorders of the hepatointestine and inflammatory reactions in the body. Abuse of alcohol can promote the destruction of the intestinal mucosal barrier, making it difficult for receptors on intestinal walls to complete signal transduction [90]. These receptors include the aromatic hydrocarbon receptor (AHR), nucleotide binding domain-like receptor protein 3 (NLRP3), peroxisome proliferator-activated receptor (PPAR), FXR, toll-like receptor (TLRs), VDR, and other receptors.

AHR is an essential ligand-activated transcription issue in the liver and intestine. It is mainly expressed in epithelial and innate immune cells and is involved in many functions, including microbial defense, cell proliferation, immune regulation, and NAD metabolism [91]. It is found that AHR is the target for regulating the homeostasis of alcohol on the hepatointestinal axis. For example, the absence of specific AHRs in the intestinal epithelial cells of mice aggravated liver injury. At the same time, the levels of Helicobacter pylori and isobutyric acid (IBA) in the intestine increased along with the expression of bacterial genes (ilvE, bkdA, and pdhD) that metabolize valine into IBA. After the administration of an AHR agonist, alcohol-induced liver injury improved, indicating that AHR may be a regulator of hepatointestinal axis homeostasis [92]. It can significantly reverse the damaged intestinal barrier, reducing insulin resistance and related inflammatory expression [93].

NLRP3 is an essential member of the inflammatory family as a component of intracellular pattern recognition in the innate immune system, which can activate the release of caspase-1 and pro-inflammatory cytokines (IL-1β and IL-18) and promote the apoptosis of inflammatory cells [94], thus increasing inflammation, steatosis, and fibrosis of the liver. Previous research has proven that activation of NLRP3 in hepatic macrophages and molecular chaperone heat shock protein 90 (HSP90) is associated with the induction of Alcoholic liver disease [95]. Choudhury and others reported that HSP90 promoted activation of NLRP3 inflammatory bodies during infection and inflammatory diseases and induced and regulated pro-inflammatory cytokines, tumor necrosis factor α, and IL-6 in Alcoholic liver disease. In addition, the expression of HSP90 and NLRP3 inflammatory body genes was positively correlated with alcoholic cirrhosis as well in a NIAAA-Gao binge mouse model [96]. More importantly, NLRP3 is the direct driving factor of IR, and it is highly expressed in NALD patients and high-fat mouse models. This is because NLRP3 directly binds and promotes the activation of protein kinase Epsilon (PKC ε), thus damaging insulin signal transduction and increasing liver steatosis [97].

PPARα and PPARγ are affiliates of the nuclear hormone receptor superfamily, expressed in adipocytes, and they can modify glucose and lipid metabolism [98]. Expression in macrophages can participate in inflammation [99]. It was found that liver fatty acid oxidation protected mice from alcohol-induced fatty degeneration. In contrast, a mouse model exposed to long-term alcohol intake showed a reverse in NIK-mediated liver fatty degeneration and fatty acid oxidation dysfunction when administered PPARα agonists [100]. Therefore, the pathological impact of NIK in ALD may be attributable to the inhibition of PPARα. Nevertheless, Nannan Sun et al. found that the promoter of peroxisome proliferator-activated receptor α (PPARα) can directly bind with KLF16 to accelerate fatty acid oxidation, reduce oxidative stress in HFD mice, and reduce lipid deposition and improve insulin resistance [101].

FXR is a ligand-activated transcription component and nuclear hormone receptor superfamily member. It is an imperative receptor for BA and lipid homeostasis regulation and can inhibit fatty degeneration and fibrosis [102]. Of course, FXR is often related to the formation and regulation of BAs. Studies have shown that FXR deficiency will change the composition of the BA pool in the serum and liver and aggravate the liver injury caused by chronic alcohol. It even includes increasing the possibility of hepatic steatosis and primary and secondary BAS levels, thus aggravating the deterioration of hepatotoxicity [103]. At the same time, Sabrina Cipriani and others’ experiments showed that the activation of FXR can effectively reduce the level of high-density lipoproteins, reduce the synthesis of free fatty acids, prevent fat deposition in the liver and muscle, and reverse insulin resistance [104].

TLRs in the liver can effectively regulate the level of inflammatory factors induced by alcoholic hepatitis [105]. With long-term drinking of alcohol, the expression of various TLR receptors such as TLR1, TLR2, TLR6, TLR7, and TLR8 in the liver increased, and the expression of related ligands for mRNA and proinflammatory factors such as TNFα, MCP-1, and iNOS increased, which is harmful to patients with hepatitis [106]. Similarly, a change in intestinal barrier function can stimulate TLR4 ligands such as endotoxins and transport them to Kupffer cells in the liver, mainly to activate nuclear factor-κB (NF-κB). Additionally, these ligands can induce the launch of proinflammatory mediators and increase the incidence of liver inflammation and fibrosis [107].

VDR helps to regulate body-related immune responses, including congenital and adaptive. Studies have shown that variations in VDR-related genes are related to the severity of liver disease, and drinking alcohol can seriously reduce the level of vitamin D in serum and enhance the synergistic effect of specific VDR haplotypes, thus accelerating the development of hepatocellular carcinoma [108]. In addition, it can form heterodimers with RXR and combine with DNA reaction elements in target genes to contribute to the biological effects of vitamin D. Of course, it was found that the activation of VDR can significantly improve the inflammatory response and steatosis induced by hepatic macrophages, as well as insulin resistance [109].

## 4. Functions of Active Ingredients in Spices in the Alcohol–Liver–Intestine Axis

Long-term alcohol intake will increase the incidence of hepatointestinal diseases. At present, drugs for clinical treatment generally include antioxidants, growth factors, anti-caspase, anti-inflammatory molecules, anti-fibrosis drugs, toll-like receptor (TLR) antagonists, and antibiotics [110]. Although it can achieve the purpose of symptomatic treatment, the side effects of drugs are inevitable. At the same time, drug therapy generally intervenes selectively according to the degree of the disease. For example, in patients with alcoholic hepatitis, abstinence and nutritional therapy are suitable for the early stage of the disease, which can improve the clinical symptoms of the disease and improve the quality of life [111]. Patients with severe alcoholic hepatitis and liver cirrhosis should be treated with drugs according to the treatment guidelines. At the end of the disease, normal drugs generally have little effect on disease intervention. Transplantation is an effective treatment to improve the survival rate, but the source of liver donors is a difficult problem [112]. In recent years, natural extracts have been regarded as substitutes for clinical drugs, which are based on special chemicals in extracts, such as silymarin, quercetin, glycyrrhizic acid, and so on. They have remarkable effects in inhibiting inflammation, accelerating alcohol metabolism, resisting oxidation, and regulating intestinal microenvironment homeostasis. As we all know, spices give food a unique flavor, and have high medicinal value. According to in vitro and in vivo experiments, the bioactive components in spices can treat and prevent diseases. This chapter expounds on the value of spices in antioxidation, lowering blood sugar, regulating intestinal flora, and strengthening the intestinal barrier from the perspective of the “alcohol–liver–intestinal axis”. Table 1 summarizes the effects of different spice extracts on liver and intestinal diseases.

### 4.1. Capsaicin

Capsaicin, a kind of “shocking” bioactive substance, is contained in the fruits of Capsicum plants. It is a derivative of vanillin, and its chemical name is 8-methyl-N-vanillyl-6-nonenamide. Capsaicin, the most common alkaloid in peppers, accounts for 70% of the active substances in these plants and is the primary source of their spicy taste. A polar amide team and a benzene ring are at the hydrophobic carbon end of its long chain; hence, capsaicin has fat-soluble characteristics and a robust, irritating smell [141]. It is reported that capsaicin can promote the secretion of acetylcholine and norepinephrine in the body, thus accelerating fat metabolism [142] and glycogen decomposition [143] and preventing the invasion of fungal pathogens [144]. In addition, capsaicin has anti-inflammatory [145], anti-cancer [146], and antibacterial functions [147], regulates the biological activity of the endocrine system [148], can reduce blood sugar and maintain stability in insulin secretion, thus slowing the development of diabetes [143], and can be used treat cardiovascular and cerebrovascular diseases [149].

As an antioxidant, capsaicin can inhibit free radicals induced by alcohol to alleviate the symptoms of hypertension, dyslipidemia, and obesity caused by oxidative stress, which may be related to the expression of cytochrome P450 2E1 (CYP2E1) and related receptors. For example, Lei Zhang and others found that taking capsaicin can significantly enhance the expression of 7α-hydroxylase and TRPV1 by reducing cholesterol through upregulation of 7α-hydroxylase expression and increasing the content of total BAs in feces [150]. More importantly, capsaicin activated TRPV1 and reduced intracellular lipid droplets induced by free fatty acids (FFAs), thus preventing the occurrence of fatty liver in vivo [151]. Moreover, in related reports, capsaicin supplementation was shown to regulate mitochondrial damage caused by alcohol, increase mitochondrial activity and function in the liver cells of a mouse model (acute alcohol feeding), increase mitochondrial GSH levels, and restore mitochondrial respiratory enzyme activity, thus inhibiting liver injury caused by alcohol [113]. Capsaicin can effectively balance matrix metalloproteinases and inhibit the expression of NF-κB. Through this means, it can eliminate the extracellular matrix (ECM) of alcoholic liver fibrosis and significantly reduce alcoholic liver injury in mice [114]. In addition, capsaicin is an agonist of the vanillin 1 (TRPV1) ion channel, which can affect membrane fluidity, ion flux, and cell ROS levels [115,152]. Studies have shown that capsaicin can counteract lipid accumulation induced by ethanol in rat livers, promote the secretion of triglycerides from the liver to plasma, and reduce the concentrations of triglycerides, cholesterol, and alcohol in serum [153]. Moreover, Ji-Hye Kang and others found that capsaicin can significantly reduce the damage to glucose tolerance and the expression of proinflammatory factors. It can even enhance fatty acid oxidation in adipose tissue and the liver, which together affect insulin resistance, mainly due to the activation of PPARγ and TRPV-1 by capsaicin [116].

Long-term intake of alcohol can increase intestinal permeability and make gram-negative bacteria and their metabolites enter the blood in large quantities, which results in significantly higher levels of plasma LPS in sufferers with ALD. Meanwhile, combining LPS and TLR4 can increase the expression of pro-inflammatory elements and aggravate the development of alcoholic hepatitis [154]. Besides, capsaicin can decrease the concentration of TNF-a, IL-6, and nitric oxide in the plasma of septic rats, reduce leakage of liver enzymes ALT and ALP into the blood, and reduce the toxic effect of LPS on liver tissue [118]. Increased intestinal permeability is frequently related to an imbalance in the intestinal flora, mainly due to an increased abundance of LPS-producing bacteria and changes in LPS biosynthesis and short-chain fatty acid (SCFA) production [155]. Interestingly, in an experiment by Kang et al., the intake of capsaicin was found to effectively increase the amount of butyric acid-producing flora (Ruminococcus and Mucor) in the intestine. This was in direct proportion to the concentration of butyrate in feces. In contrast, capsaicin leads to a limit in the degrees of S24_7 and PAMP, which are contributors to the LPS family and weaken the TLR4 pro-inflammatory reaction [119]. As we all know, capsaicin can activate TRPV1, an extended member of endogenous cannabinoid (income). Suppose mice lacking cannabinoid 1 receptor (CB1R) in intestinal epithelial cells show the aggravation of intestinal barrier dysfunction induced by diet. In that case, studies have shown that capsaicin can affect the expression of CB1R to reduce intestinal barrier damage and endotoxin production. To sum up, capsaicin can effectively prevent microbial imbalance, intestinal barrier dysfunction, and chronic inflammation caused by alcohol or another dietary intake [119].

### 4.2. Allicin

Garlic is a common substance in spices. Garlic juice contains a bioactive substance, allicin, whose chemical name is diallyl thiosulfinate. There are two substances in different compartments of garlic cloves, namely alliin and alliinase, when chopped, crushed, chewed, and mixed evenly, thus producing a significant amount of allicin in less than 6 s [156]. The sulfur atoms in allicin are highly active. They can be easily decomposed to produce secondary organic sulfur compounds under certain processing and storage conditions, such as a specific concentration, pH value, and temperature [157]. Allicin has broad-spectrum antibacterial activity and can inhibit various drug-resistant bacterial strains, including Escherichia, Staphylococcus, Proteus, Pseudomonas, and Enterococcus [158]. Allicin is also widely used in antifungal studies and can effectively inhibit *Candida* and *Aspergillus* [159]. In addition, allicin has the effects of lowering blood pressure [160], resisting cancer [161], improving antioxidant activity [162], improving insulin resistance [163], and alleviating steatosis [164].

Long-term alcohol consumption can lead to the disease of fat metabolism, which manifests as a decrease in mitochondrial lipid peroxidation and an increase in triglyceride synthesis. Simultaneously, TNF-α upregulates (sterol regulatory element-binding protein 1) SREBP-1, aggravating the liver disease [165]. Studies have shown that allicin can significantly improve the effects of fatty metabolism disorder and alcoholic fatty liver disease (AFLD) symptoms in mice by reducing the levels of aspartate aminotransferase, alanine aminotransferase, and triglyceride in the liver and the relative weight of the liver [120]. Long-term alcohol intake will increase the ROS level, increasing the reactive oxygen species and pro-inflammatory cytokines (TNF-α, IL-1β, and IL-6) in AFLD patients’ livers compared with ordinary people. Nevertheless, taking allicin can reduce the initiation of ROS and the expression of the CYP2E1 protein, thus reducing oxidative stress. At the same time, allicin can significantly reduce the inflammatory reaction of the liver by inhibiting the activation of the inflammatory body of NLRP3 and reducing the expression of caspase-1 and the secretion of IL-1β, IL-18, IL-6, and TNF-α [121]. Studies have shown that the oxidation balance in the liver of AFLD patients is dysregulated, which leads to a high level of glutathione catalase (GPx), with liver glutathione (GSH) transformed into glutathione disulfide (GSSH) under the action of GPx. Allicin can increase the degrees of GSH and catalase (CAT) in the liver to some extent, thus alleviating liver injury in patients with AFLD [120]. In addition, allicin can induce the up-regulation of ABCA1, promote cholesterol excretion, reduce lipid accumulation through PPARγ/LXRα signal transduction, and reduce alcohol damage to the liver [166]. At the same time, allicin can reduce serum glucose and insulin levels and reduce triglyceride and digestive tract fatty acid absorption, thereby improving insulin resistance and impaired glucose tolerance [122].

In recent years, many studies have pointed out the liver protection advantages of allicin from the perspective of its molecular action, and a few studies have focused on the role of allicin and intestinal flora. This may be because allicin has specific antibiotic characteristics [167]. However, it differs from antibiotics in the market, mainly because allicin is unstable and metabolized into other bioactive substances such as diallyl sulfide, diallyl disulfide, and diallyl trisulfide [168]. Based on the functional characteristics of intestinal flora, whether allicin intake can restore the microecological balance has become a topic of interest for researchers. Panyod and others found that allicin can effectively inhibit the formation of TMAO, enhance the variety of intestinal microbial flora, and enlarge the relative abundance of advisable organisms after administering garlic juice for 1 week [169]. Moreover, Gao et al. used an IPEC-J2 cell monolayer to simulate the intestinal barrier and verified through experiments that allicin has an intervention impact on LPS-induced intestinal epithelial barrier injury, which was mainly due to allicin’s extent in transepithelial resistance (TEER), reduction in paracellular permeability, and enhancement of zona clostridium 1 (ZO-1) integrity of the IPEC-J2 cell monolayer, as well as preventing LPS-induced activation of the Nrf2/HO-1 pathway-dependent antioxidant system [170]. In addition, the study also found that the F/B ratio in the intestine of AFLD mice was significantly reduced after the administration of allicin [124]. Furthermore, AGMT (intestinal flora induced by allicin transplanted into mice with HFD) significantly increased Blautia (microbial population producing SCFA) and Bifidobacterium numbers in HFD mice and improved SCFA in the cecum [125]. Most experiments have shown that allicin does not disturb the intestinal microecological balance and has specific pharmacological value for AFLD and hepatointestinal complications.

### 4.3. Curcumin

Curcumin is a nutritional and dietary polyphenol derived from the tuber Curcuma longa. Its chemical name is difuranformylmethane, and it is among the most widely studied substances among the active ingredients of spices. Curcumin displays vigorous anti-inflammatory activity and can effectively intervene in inflammatory diseases [171]. It can also regulate cardiovascular diseases, improve endothelial function, and promote heart and vascular internal membrane health [172]. Curcumin has a significant effect on cancer, helping to slow the spread of tumor cells and forestall the formation of tumors [173]. Curcumin can also promote glucose and lipid metabolism, enhance the body’s antioxidant activity, and has specific therapeutic and preventive effects on diabetes [174]. In addition, the functional characteristics of curcumin include preventing Alzheimer’s disease [175], treating depression [176], improving skin health [177], and fighting free radicals [178].

The synthesis of fatty acids and unsaturated fatty acids is positively correlated with ethanol-induced hepatic steatosis, in which ethanol can increase the contents of stearic acid, oleic acid, and linoleic acid, thereby inducing fatty acid ethyl ester, fatty liver, and alcoholic liver injury. Experiments have shown that curcumin intake can interfere with the synthesis of fatty acids and unsaturated fatty acids and forestall the progress of fatty liver [126]. In addition, curcumin can regulate stearic acid, oleic acid, and linoleic acid and drastically inhibit the manufacturing of fatty acid synthase. Nevertheless, curcumin showed a potential inhibitory effect on the synthesis of triglycerides and very low-density lipoproteins in the liver and prevented chronic alcohol from inducing and destroying the liver. At the same time, curcumin can alleviate liver damage precipitated through continual ethanol [127]. It is well known that in the presence of ω-3 PUFA, ethanol induces CYP2E1 to produce more ROS, which leads to oxidative stress. Curcumin treatment can increase the activity of PON1 and the value of -SH in the liver while reducing the oxidative sensitivity of LDL [179]. Alcohol intake can promote the production of pro-inflammatory markers and apoptosis in hepatocytes [180]. Among these effects, the expression of related genes mediated by NF-κB is often linked to the pathogenesis of ALD. Curcumin can effectively block the activation of NF-κB mediated by endotoxin and inhibit the expression of cytokines, chemokines, COX-2, and iNOS in Kupffer cells, preventing ALD [181]. A study by Lu et al. demonstrated that curcumin reduces alcohol-induced hepatocyte necrosis by activating nuclear issue erythroid 2-related factor 2 (Nrf2) [128]. Surprisingly, curcumin reduces endoplasmic reticulum stress through the cAMP/PKA pathway and reduces FFA inflow into the liver by blocking FFA transport, thus improving insulin sensitivity and inhibiting glucose production [129].

Interactions of the intestinal flora mediate the pharmacological activities of curcumin; on the one hand, curcumin directly regulates intestinal microflora, and on the other hand, intestinal microflora biotransforms curcumin and produces active metabolites [182]. An experiment by Feng et al. confirmed that curcumin significantly changed the structure of the intestinal microflora and changed microbial diversity at different classification levels, such as significantly reducing the number of organisms positively related to obesity in mice. Curcumin also increased the number of species of Gordon’s bacteria in the intestine; curcumin administration facilitated an addition of 39 OTUs, among which members of seven OTUs may produce SCFAs [130]. In addition to changes in the intestinal flora, the protection of the intestinal barrier is also an evaluation index. In Wang et al.’s experiment, curcumin was found to act on the intestinal epithelium and intestinal barrier mainly. Furthermore, curcumin intake can significantly limit the destruction of the intestinal epithelial barrier induced by LPS, reduce the secretion of IL-1β and activation of p38 MAPK induced by IL-1β, increase phosphorylation of tight junction proteins, and destroy its standard arrangement [183]. Similarly, in an experiment by Liu et al., curcumin supplementation was shown to alleviate inflammatory reactions in rats with heterogenous sepsis, upregulate expression of intestinal tight junction proteins ZO-1, occludin, and claudin-1, protect intestinal barrier function, and regulate the ERK/JNK signaling pathway to inhibit activation and apoptosis due to NF-κB p65 [184]. The outstanding performance of curcumin in antioxidant, anti-inflammatory, antibacterial, and hypoglycemic effects provides clinical ideas for improving alcoholic hepatointestinal diseases.

### 4.4. Cinnamine or Cinnamic Acid

Cinnamon bark is a common spice in stewed food. It can also be used to extract cinnamon oil and synthesize cinnamic acid. There are two main active substances in cinnamon bark: cinnamic acid and cinnamaldehyde. The latter mainly oxidizes the former. Cinnamaldehyde and cinnamic acid both have antibacterial [185], anti-inflammatory [186], and anti-oxidation [187] properties, and can regulate sugar and lipid metabolism in the liver [188]. In recent years, researchers have treated and prevented ALD and its complications with active substances in cinnamon.

Cinnamic acid is an active phenolic acid in plant food with many functional characteristics that can protect the liver. In an experiment by Yan et al., providing a specific dose of cinnamic acid to a mouse model with alcoholic hepatitis toxicity led to reduced gene expression of CYP2E1, p47phox, gp91phox, COX-2, and NF-kB induced by ethanol. It also enhanced the expression of Nrf2 in the cytoplasm and nucleus because p47phox inhibits the ubiquitination of Nrf2 and activates NRF 2. In addition, cinnamic acid can interfere with alcohol-induced oxidative stress, reducing the release of ROS, oxidized glutathione (GSSG), IL-6, TNF-α, and prostaglandin E 2 (PGE 2) in the liver, thereby reducing alcohol-induced hepatotoxicity [132]. Moreover, cinnamic acid reduces inflammatory cell infiltration into liver tissue. Alcohol can aggravate the fatty degeneration of the liver, leading to the secretion of a giant variety of chemokines, recruitment of inflammatory cells to infiltrate into adipose tissue and liver, transformation of these cells into inflammatory macrophages, and activation and release of proinflammatory factors. Interestingly, Lee and others found that cinnamic acid can reduce complications caused by obesity, such as glucose tolerance, dyslipidemia, and fat deposition, without showing hepatorenal toxicity. Further, cinnamic acid can reduce the number of Ly6c+ monocytes and M1 macrophages, expression of TNF-α induced by steatosis, and infiltration of macrophages in liver tissue [133]. At the same time, Da-Wei Huang and others found that cinnamic acid can promote tyrosine phosphorylation of insulin receptors, up-regulate the expression of insulin signal-related proteins, including insulin receptor, phosphatidylinositol 3 kinase (PI3K), glycogen synthase (GS) and glucose transporter 2 (GLUT2), and increase glucose uptake, thus reducing insulin resistance in cells [189].

Based on the antibacterial and anti-inflammatory activities of cinnamic acid and cinnamaldehyde, their effects on the intestinal flora are often studied. For example, Jiang et al. used a clomipramine-induced STC model in mice and cinnamic acid as the intervention medium. Their research found that, first, cinnamic acid can significantly improve the diversity and richness of beneficial microbial communities, such as increasing the composition and quantity of Firmicutes, Verrucous Microorganisms, Lactococcus, Ackermann, Lactuccia, and Acinetobacter; second, cinnamic acid also significantly promoted the production of single-chain fatty acids such as acetic acid, butyric acid, propionic acid, and valeric acid [134]. Since cinnamaldehyde is easily oxidized in the outside world, it must be combined with microcapsule technology to improve stability and prevent environmental pressure. In Xiao et al.’s research, microencapsulated cinnamaldehyde increased the antioxidant potential of the liver, duodenum, and colon. In addition, 16S rRNA gene sequencing statistics showed that microencapsulated cinnamaldehyde considerably regulated the intestinal flora and its metabolites, increasing the abundance of Bacteroides, Bacteroides/Cladosporium, unclassified Lactobacillus, Lactobacillus, and Brautia, and decreasing the abundance of Lactococcus, UCG-014, fecal, and Brautia [190]. Considering the low toxicity and high biological activity of cinnamic acid and cinnamon, it is necessary to make them dietary supplements for patients with alcoholic hepatointestinal diseases.

### 4.5. Geraniol

Fragrant leaves, thyme, and citronella are a kind of common spices. Their aromatic properties are mainly attributable to the volatility of geraniol. In recent years, many researchers have centered on the active function of geraniol in the body, that is, its antibacterial [191] and anti-inflammatory [192] functions, improvement of oxidation resistance [192], potential to treat Parkinson’s disease [193], promotion of liver regeneration [194], and potential anti-tumor properties [195]. Geranyl is a group derived from vanillin by the removal of hydroxyl, with geranyl pyrophosphate (GPP) and geranyl pyrophosphate (GGPP) among the leading derivatives of geranyl. These compounds are also essential intermediates in cholesterol biosynthesis.

Long-term drinking will lead to a significant decrease in total protein and its components, albumin and globulin. Even the liver cells are distorted, accompanied by a significant increase in oxidative stress parameters (MDA and H2O2). The experiments of Samah A et al. showed that the ethanol extract of Thyme leaves could effectively interfere with alcohol-induced hepatotoxicity in rats, such as reducing the activities of alkaline phosphatase (ALP), aspartate aminotransferase (AST) and alanine aminotransferase (ALT), increasing the production of total protein, albumin and globulin, and even improving the oxidative stress response and protecting the liver [136]. At the same time, the same result was obtained in the research of Hasan et al.; geraniol has been proven to effectively decrease the levels of lipid peroxidation and serum toxicity markers (AST, ALT, LDH) while increasing the activities of catalase, glutathione peroxidase (GPxs), glutathione reductase (GR), superoxide dismutase (SODs), glutathione, and related enzymes, thus significantly improving 2-AAF-induced oxidation. At the same time, Geraniol treatment can also downregulate the caspase-3,9, COX-2, NFkB, PCNA, iNOS, and VEGF expression [196]. Surprisingly, geraniol was shown to maintain mitochondrial function in patients with steatohepatitis, reducing mitochondrial formation, enhancing adenosine triphosphate formation and membrane integrity, and restoring mitochondrial electron transport chain enzyme activity. It also decreased the expression of uncoupling protein 2 in the liver and enhanced the mRNA expression of PPAR and the function of carnitine palmitoyl transferase I (CPTI) in mitochondria. Geraniol can also reduce the formation of malondialdehyde (MDA) and 3-nitrotyrosine in the liver, enhance the function of glutathione-S-transferase (GST), and downregulate the expression of INOS and CYP2E1 to reduce oxidative stress in the liver [137]. Interestingly, geraniol can reduce TNF, IL-1, and NF-κB levels and the formation of MDA induced by DENA, as well as increase the activities of GSH and antioxidant enzymes in the liver, consequently having a defensive effect on liver cancer [197]. Moreover, research by Mohammed shows that geraniol may protect the liver by inducing activation of MAPK, p38, and JNK, reducing expression of the PPAR-γ protein, and reversing inflammatory and oxidative stress reactions in the liver [138]. To address considerations of dose and hepatotoxicity, Pavan and others found that mice taking a hundred and twenty mg/kg geraniol for four weeks confirmed enhanced antioxidant defense ability and no signs of hepatotoxicity [198].

Based on its hydrophobicity, geraniol effectively binds to bacterial cell walls, alters their dynamic organization, and leads to ion loss and ATP depletion. The antimicrobial activity of geraniol does not appear to have a specific target and may attack beneficial flora in addition to killing pathogenic bacteria in the human gut. However, studies have shown that human pathogenic organisms are more sensitive to geraniol than commensal species, which indicates a possible mechanism underlying geraniol’s intestinal action. Ricci et al. conducted a double-blind, randomized, managed trial via fifty-six patients with inflammatory bowel syndrome (IBS), identified following the Rome III criteria, and found that remission of IBS in vivo used to be notably higher in patients handled with geraniol and that the treated vs. untreated patients showed significant differences in the composition of their intestine microbiota. In particular, there was a giant minimizing effect in Oscillospira, a genus of the Lactococcaceae family (*p* = 0.01), and a trend towards a decrease in the Danubacteriaceae and Clostridiaceae families (*p* = 0.1) was observed. In contrast, a trend towards an increase was observed in the other Lactococcaceae taxa, particularly in fecal *E. coli* (*p* = 0.09) [199]. Thapa et al. also showed that 500 ppm thymol and geraniol could inhibit total bacteria and reduce the number of Clostridium difficile. Previous experimental data showed that about 100 ppm of thymol and geraniol could effectively inhibit pathogens in the small gut without worrying about diminishing beneficial symbiotic colon organisms in the distal intestine [200]. In addition, geraniol has the effect of downregulating MRP2, which can entirely and partially offset the increase in intestinal IL-1β and IL-6 levels caused by fructose, even reducing the activities of lipid peroxide products and superoxide dismutase and restoring the ratio of glutathione reduction to glutathione oxidation, which improves the redox imbalance in the intestinal tract [139]. In summary, geraniol has been shown to improve the function of the hepatic-gut axis in terms of antioxidant, anti-inflammatory, lipid modulation, and intestinal flora alteration properties.

## 5. Conclusions

In modern society, overeating (excessive drinking or high-calorie intake) has become the root cause of various diseases. Among them, metabolic diseases caused by drinking are the main focus of attention. Generally speaking, the time of alcohol intake is closely related to the development of the disease. This process often involves many factors, including the activity of alcohol-metabolizing enzymes in the liver, the species distribution of intestinal flora, the integrity of the intestinal barrier, and tissue inflammation. However, we still lack enough information to grasp the healthy crosstalk between the alcohol–liver–intestine axis. This uncertainty provides a hypothesis for us to explore the molecular interaction mechanism further in the future. First, in vivo and in vitro combinations can effectively evaluate alcohol–liver–intestinal health. Although the experiment from cells to mice can not fully meet the clinical needs, it is still the first step for scholars to explore the mechanism of a substance. For the bioactive substances and drugs that have not been studied, it is still necessary to carry out cell tests in order to master the toxicity and mechanism of action of the active substances. Secondly, for the mouse model, the amount and frequency of alcohol use are significant. Besides the changes in body weight and blood sugar, organ diseases and biochemical analysis can often be combined with cell experiments, so we must conduct related experiments for different test purposes and properties.

With people’s increasing attention to diet and health, spices have been continuously explored in many studies as a typical representative of the homology of medicine and food. As mentioned in the article, the characteristics of its intervention on alcohol-induced hepatointestinal diseases are as follows: First, the bioactive factors in spices can exert antioxidant activity in the hepatointestinal environment and reduce the oxidative stress response caused by diseases. Second, these factors can interfere with fatty acid synthesis, promote glucose and lipid metabolism, and alleviate liver damage caused by steatosis. Thirdly, they can effectively restore the intestinal microenvironment by regulating the balance of intestinal flora, promoting the production of SCFAs, and even enhancing the intestinal barrier function. However, the application of spice bioactive substances in alcohol-induced hepatointestinal diseases is still limited, including the influence of alcohol metabolizing enzymes in the liver, the difference of spice activity before and after digestion, and the crosstalk between spice bioactive substances–alcohol–intestinal flora. Therefore, in future research, we must design more systematic experimental models and clinical trials to verify better the relationship between spice active substances and intestinal microbial groups, metabolic groups, and immune groups. It helps us to identify specific spice-active substances to intervene in the intestinal flora. At the same time, it can effectively alleviate the problems of hepatointestinal diseases caused by alcohol by combining with various omics technologies. Of course, precision medicine is gradually becoming the future treatment trend, and the development of spice-active substances has a high prospect based on improving people’s attention to precision nutrition. These works laid a foundation for the intervention of spice-active substances in hepatointestinal diseases caused by alcohol.

## Figures and Tables

**Figure 1 ijms-25-03728-f001:**
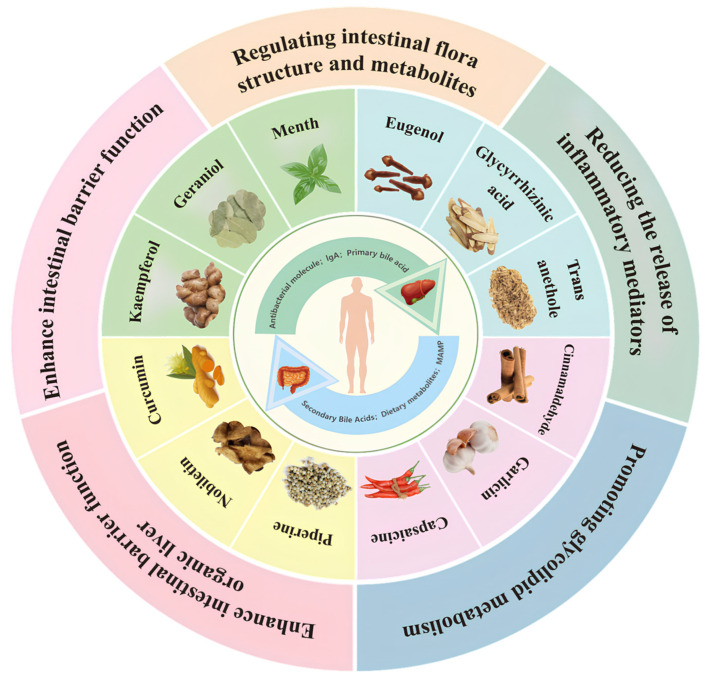
Regulatory mechanisms and potential value of active substances in 12 aromatic spices as interventions in alcoholic liver disease and liver and intestinal disorders.

**Figure 2 ijms-25-03728-f002:**
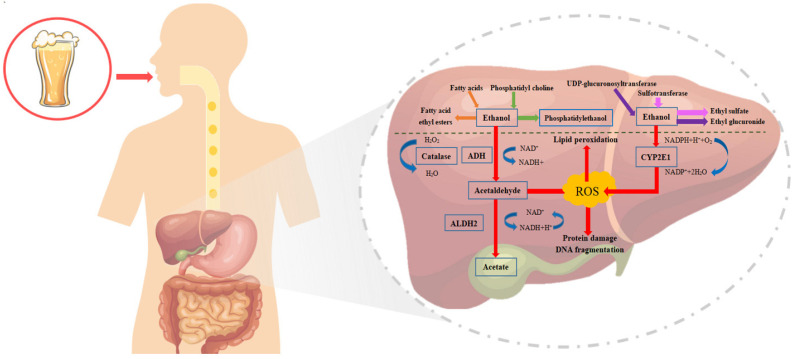
The two different metabolic pathways of alcohol in the liver, oxidative and non-oxidative. Alcohol is converted to acetaldehyde, a hepatotoxic compound, by the oxidation of ADH, CYP2E1, and catalase, and also produces large amounts of ROS. In the non-oxidative pathway, alcohol is combined with endogenous metabolites. Various enzymes are produced, including FAEE, PEth, EtG, and EtS.

**Figure 3 ijms-25-03728-f003:**
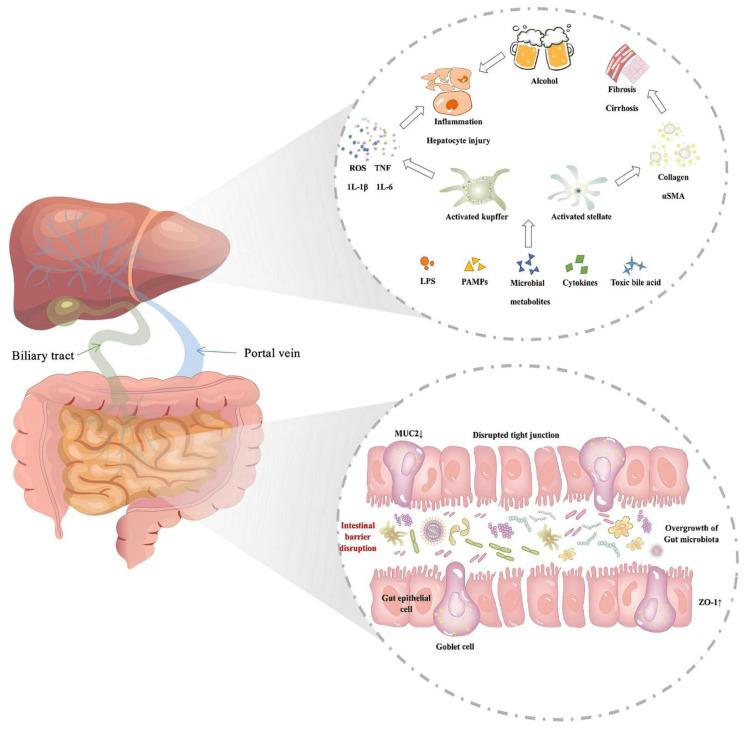
Pathophysiology of the enterohepatic circulation imbalance in alcoholic liver disease. Long-term intake of alcohol can lead to changes in intestinal permeability and the intestinal microflora, leading to increased secretion of inflammatory cytokines.

**Figure 4 ijms-25-03728-f004:**
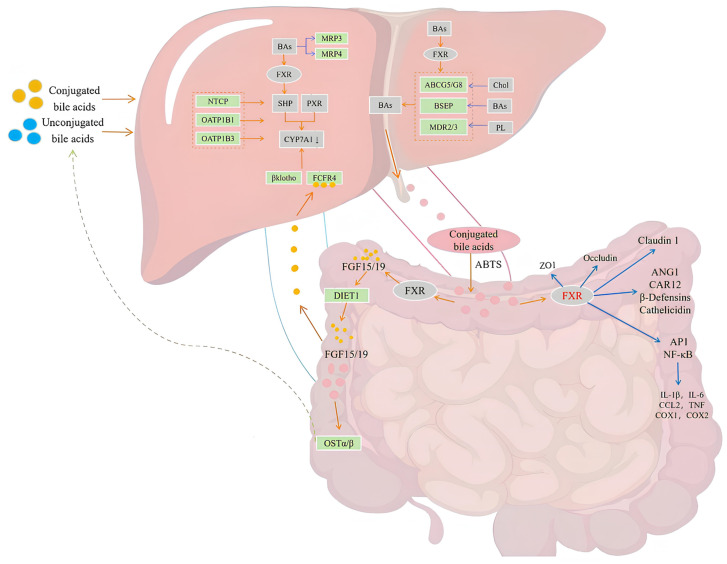
The dual function of FXR in the liver and intestine is mainly manifested in the transport and inhibition of bile acids. In hepatocytes, FXR can inhibit Cholesterol 7a-hydroxylase (CYP7A1) transcription and BAs synthesis by inducing SHP and can also promote BAs secretion by inducing bile salt export pump (BSEP). FXR induces fibroblast growth factor 15/19 (FGF15/19) in the intestine. Under the action of DIET1, FGF15/19 enters the portal vein blood and circulates to the liver. FGF4/β klotho binds to FGF15/19 to inhibit the synthesis of BA. Interestingly, FXR can induce hetero-organic solute transporter-α/β (OST α/β) and promote the entry of binding bile acids and non-binding bile acids into hepatocytes. At the same time, FXR can also maintain intestinal homeostasis and the integrity of the intestinal epithelial barrier and even inhibit the transcription of proinflammatory factors.

**Table 1 ijms-25-03728-t001:** Function of active ingredients in spices in liver and intestine ^a^.

Category	Active Substance, Administration Dose and Period	Effects	Ref.
Capsicum Frutescens Extract	0.01% capsaicin (15 mg/kg b. wt/d), gavage, 24 weeks.	↑TRPV1, Phospho-HSL, CPT-1, PPARδ, LC3-II, Beclin1, Atg5, Atg7 ↓FFAs, Hepatic enzymes, Inflammatory factor	[113]
	Capsaicin (10 or 20 mg/kg b. wt/d), gavage, 10 days.	↑MMP/TIMP balance, Mitochondrial respiratory enzyme activities, GSH ↓CYP2E1, ROS, NF-κB (p65), Lipid peroxidation	[114]
	0.075% capsaicin (100 mg/kg b. wt/d), gavage, 8 weeks.	↑Adiponectin level, AMPK, CPT-1, CD36, ↓β-oxidation, Fat accumulation, ACC, FAS, PEPCK-C, G6Pase, AST, ALT	[115]
	0.015% capsaicin, gavage, 10–20 weeks.	↑PGC-1α, TRPV-1 ↓TNFα, MCP-1, IL-6, Leptin, Insulin resistance	[116]
	Dihydrocapsiate (100 mg/kg b. wt), gavage, 12 weeks.	↑UCP1, PGC1α, COX7a, TMEM26, ACOX1, TRPV-1, SCFAs ↓PLIN1, MCP1, LPS, NO, Insulin resistance, SREBP1, FABP4, ADI POQ, LEPTIN, TNFα, ROS, PEPCK, G6Pase	[117]
	Capsaicin (15, 150 and 1500 μg/kg), oral and intraperi gavage, 4 h.	↑MDA, GSH, TRPV-1 ↓ALT, AST, ALP, NO, TNF-α	[118]
	0.01% capsaicin, gavage, 12 weeks	↑Butyrate-producing bacteria, SCFAs ↓CB_1_ expression, Proinflammatory cytokines, gut permeability, LPS	[119]
Garlic Extract	Garlicin (5 and 20 mg/kg b. wt/d), gavage, 4 weeks.	↑GSH, CAT, SOD ↓TNF-α, IL-1β, IL-6, AST, ALT, Hepatic triacylglycerol, SREBP-1	[120]
	Garlicin (25 and 50 mg/kg b. wt/d), gavage, 4 weeks.	↓ROS, CYP2E1, GRP78, CHOP proteins levels, p-IRE1α, p-ASK, TRAF2, MAPK/NF-κB/NLRP3 signal pathways	[121]
	Garlic extract (4% w/w), gavage, 7 weeks.	↑Bifidobacterium, Clostridium cluster XVIII, Prevotella ↓Insulin resistance, TG	[122]
	Garlic oil (20 and 40 mg/kg/d), gavage, 10 weeks.	↑SOD, GSH-Px, Sirt1, PGC-1α, ZO-1, Claudin1 proteins ↓Triglyceride levels, MDA, FoxO1	[123]
	Garlicin (5 and 20 mg/kg/d), gavage, 4 weeks.	↑ALDH ↓LPS, CD14, TLR4, TNF-α, IL-1β, IL-6, F/B ratio	[124]
	1% m/m allicin, gavage, 13 weeks.	↑F/B ratio, SCFAs, IL4, IL10, IL13 ↓Insulin resistance, IL6, IL1β, IL16, MCP1, TNFα	[125]
Turmeric Extract	Curcumin (60 mg/kg b. wt), gavage, 4 weeks.	↓Fatty acids synthesis, Biosynthesis of unsaturated fatty acids, Fatty acid synthase, ALT, Steatosis	[126]
	Curcumin (150 mg/kg b. wt/d) gavage, 8 weeks.	↑PON1, HTLase, ↓Triglyceride, VLDL, ALT, AST, HDL-C, ROS	[127]
	Curcumin (100, 200 and 400 mg/kg/d), gavage, 4 weeks.	↑Nrf2/P53 ↓HMGB1, RIP3, MLKL, JNK	[128]
	Curcumin (50 mg/kg), gavage, 10 days.	↑AMP, AMPK ↓NF-κB, cAMP, DAG, PKCε, Insulin resistance	[129]
	Curcumin (200 mg/kg/d) gavage, 4 weeks.	↑ZO-1, Occludin, Allobaculum, Bacteroides, Blautia, Phascolarctobacterium, SCFAs ↓ALT, AST, NF-κB, TLR4, LPS	[130]
	Curcumin (75 and 150 mg/kg/d) gavage, 4 weeks.	↑MMP, ATPase activities, SOD, GSH-Px ↓MPTP, MDA, PGC-1α, NRF-1, Mn-SOD, GRP78, NF-κB, PERK, IRE1α, IκBα, TNF-α, IL-1β, IL-6	[131]
Cinnamon Extract	Fennel Cinnamic acid (40 or 80 mg/kg b. wt/d), gavage, 5 days.	↑GSH, GPX, Catalase activities, Nrf2, ↓ADH, CYP2E1, ROS, GSSG, IL-6, TNF-α, PGE2, NO, NOS, COX-2, iNOS, COX-2, NF-κB p65	[132]
	Cinnamic acid (100 and 200 mg/kg/d), gavage, 8 weeks.	↑HDL-C ↓TNF-α, IL-1β, IL-18, Insulin resistance, Ly6c, LDL-C	[133]
	Cinnamic acid (100 and 200 mg/kg/d), gavage, 4 weeks.	↑Firmicutes’ composition and abundance, AA, BA, VA, 5-HT ↓VIP	[134]
	Cinnamaldehyde (100 and 200 mg/kg b. wt/d), gavage, 29 days.	↑LC3II/I radio, ↓Triacylglycerol, Insulin resistance, Srebp1c, Acaca, IRE1α, EIF2α	[135]
Passion Leaf Extract	Thymus vulgaris leaves alcoholic extract (500 mg/kg/d), gavage, 21 days.	↑SOD, CAT, GR, GST, GPx, GSH, HDL-C ↓AST, ALT, MDA, LDL-C	[136]
	Geraniol (200 mg/kg/d), gavage, 10 weeks.	↑CPT-I, PPARα ↓Caspase-9, Caspase-3, ROS, UCP2, CYP2E1, iNOS, MDA, 3-NT, TNFα, IL-6	[137]
	Geraniol (100 and 200 mg/kg/d), gavage, 1 week.	↑DENA ↓ALP, TBL, GGT, DENA, CCl4, LPO, ODC, TNF-α, IL-1β, NF-κB	[137]
	Geraniol (100 and 200 mg/kg b. wt/d), gavage, 10 days.	↑PPAR-γ, ↓P38-MAPK, JNK, ALT, AST, ALP, NF-κB, TNFα, IL-6, COX-2, iNOS	[138]
	Geraniol (250 mg/kg/d), gavage, 14 days.	↓TGA, IL-1β, IL-6, Mrp2, TNF-α,	[139]
	Thymus quinquecostatus Celak extract (60 mg/kg/d), gavage, 6 weeks.	↑The abundance of Firmicutes, Bacteroidetes, Proteobacteria, Parabacteroides, Bacteroides, Peptococcus, Muribaculum, Tyzzerella, Ruminococcaceae UCG-013, Leuconostoc, ↓LPS, TLR4, ROS	[140]

^a^ Note: PON1, paraoxonase 1; HTLase, homocysteine thiolactonase; HMGB1, High mobility group box 1; RIP3, Receptor-interacting protein 3; MLKL, mixed lineage kinase domain-like pseudokinase; JNK, p-c-Jun N-terminal kinase; AMP, Adenosine monophosphate; AMPK, Adenosine 5′-monophosphate (AMP)-activated protein kinase; PKCε, Protein Kinase C epsilon; MMP, Matrix metalloproteinases; NRF-1, nuclear respiratory factor 1; GPX, glutathione peroxidase; iNOS, inducible nitric oxide synthase; COX-2, cyclooxygenase-2; VIP, Vasoactive Intestinal Peptide; EIF2α, eukaryotic initiation factor 2 alpha; CPT-I, Carnitine palmitoyltransferase-I; UCP2, Uncoupling protein 2; 3-NT, 3-nitrotyrosine; DENA, diethylnitrosamine; TBL, Transducin beta-like; GGT, γ-glutamyl transferase; ODC, ornithine decarboxylase; Mrp2, ABC transporters multidrug resistance associated protein 2.

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
