# Peer review of "Regulation Mechanism and Potential Value of Active Substances in Spices in Alcohol–Liver–Intestine Axis Health"

_ijms, 2024, doi:10.3390/ijms25073728_

Round 1

Reviewer 1 Report (Previous Reviewer 2)

Comments and Suggestions for Authors

The article entitled “Regulation mechanism and potential value of active substances 2 in spices in alcohol-liver-intestine axis health” is a review specifically focused on the metabolic transformation of alcohol in the liver and intestine and summarizes the potential value of various perfume active substances in improving liver and intestine diseases caused by alcohol. The article focused on the spices for liver disorders. Following are a few points need which might improve the article a bit more:  

1.       The Introduction is informative but the actual novelty of the article is not clear in the introduction section. Authors should mention the aim and objective of the article. Moreover why only the natural spices are selected for the review articles is also not very clear in the introduction section

2.       How much of the alcohol can alter the BA balance? Authors should provide detailed reasons for the alteration of the BA and the correlation of the BA pathways with liver cancer due to alcohol consumption.

3.       How alcohol consumption increases the fungal species in the intestine. Authors need to explain the reasons in detail and also explain the PAMPs for the intestinal flora

4. The authors provided a brief detail about the current state of drug treatment for alcoholic liver disease and why the natural compounds are more beneficial than the chemical drugs.

5.       Provide clinical evidence of the natural products for the ALD.

6.       In the conclusion authors focused on the drug delivery system for the Natural compounds for ALD. To support their statement authors should provide a brief of the different formulation strategies for the natural compounds.  

7.       Authors should do a thorough grammar check of the article

Comments on the Quality of English Language

Moderate grammatical editing is needed

Author Response

Reviewer 2 Report (New Reviewer)

Comments and Suggestions for Authors

The paper focuses on the potential therapeutic role of spice-derived bioactive substances in mitigating hepatointestinal diseases induced by alcohol. The findings suggest that these substances exhibit antioxidant properties, reducing oxidative stress in the liver and intestine. Furthermore, they can interfere with fatty acid synthesis, enhance sugar and lipid metabolism, and alleviate liver injury associated with steatosis. Notably, the study underscores the effective regulation of intestinal flora, promoting short-chain fatty acid production and restoring the intestinal microenvironment. Overall, the article sheds light on the promising medicinal value of spice compounds in addressing alcohol-induced liver and intestine-related disorders.

I would like to kindly ask you to explain what the authors had in mind when posting Table 1 and why when describing the substances in the subsections. In my opinion, there is no link between the substances shown in the table and the substances described in subsections 4.1 etc.

While the article rightly underscores the potential of spice-active substances, the language should remain cautious, avoiding overly optimistic terms that may oversell the current understanding of the subject.

Round 2

Reviewer 1 Report (Previous Reviewer 2)

Comments and Suggestions for Authors

Dear Authors, 

The article is very well revised and can be accepted in its current form. 

Comments on the Quality of English Language

Moderate editing of English language required

Author Response

感谢您的批评和指导,因为在您的帮助下,文章的结构和内容更具可读性和参考性。

This manuscript is a resubmission of an earlier submission. The following is a list of the peer review reports and author responses from that submission.

Round 1

Reviewer 1 Report

Comments and Suggestions for Authors

Huang's review aims to elucidate the mechanisms underlying the beneficial effects of spices in hepatic and intestinal diseases. However, there are significant concerns regarding the manuscript's structure and focus. The presented information is abundant but lacks clear organization and alignment with the stated title, resulting in the absence of definitive objectives.

A primary concern is the breadth of pathologies discussed. The review appears to cover intestinal pathologies, alcoholic liver disease (ALD), and non-alcoholic fatty liver disease (NAFLD), which is overly broad. There is a need for more precise objectives focusing on specific diseases.

The connection between spices and these diseases is underdeveloped, particularly at the outset of the manuscript. The initial discussion on bile acids, for instance, seems disconnected from the central theme, leaving the reader perplexed.

There is also a need for caution when linking certain mechanisms to hepatic diseases, especially NAFLD. The associations may relate more to insulin resistance rather than directly to hepatic pathology. I suggest that the authors expand their literature review to reflect this perspective.

Additionally, the figures included in the manuscript appear to be non-original, derived from other literature sources, and lack sufficient resolution, rendering them unreadable. This aspect further detracts from the clarity and effectiveness of the manuscript

Comments on the Quality of English Language

Must me improved

Reviewer 2 Report

Comments and Suggestions for Authors

Dear Authors,

The article titled “Regulatory mechanisms and potential value of active substances in spices in ameliorating alcoholic liver disease and liver and intestinal disorders " is undeniably intriguing. Below are a few suggestions for improving the article.

1.     The introduction is very informative and clear with the idea. The details of the alcohol is interesting but actual causes of the liver damage by the alcohol is not clear.  Authors should highlight the primary mechanism of the liver damage by alcohol in the introduction for the study.   

2.     Provide a high-resolution figure 1, 3.

3.     The nuclear receptors and membrane 183 G protein-coupled receptors are one of the most significant for the BA transport in the body. Authors need to explain the mechanism pf action in more details with some pathway images for better understanding.  

4.     Authors specifically focused on the natural resources. Are there any chemical / drugs also in use for the ALD. Authors can briefly highlighted the therapeutic approaches of ALD.  

Comments on the Quality of English Language

English needs minor spelling corrections. 

Round 2

Reviewer 1 Report

Comments and Suggestions for Authors

Dear Authors, thank you for your new manuscript. But your responses to my first comments (3 and 4) in my first report are essential and you didn't respond sufficiently, it is the same issue for my comment number 1 where the argument and changes are very minimal. Furthermore, when you write a letter of reply to the reviewers, please indicate precisely and show where you have made the changes in the manuscript. Your manuscript is very long and it's important to help the reviewer see your improvements.  

Comments on the Quality of English Language

.

Reviewer 2 Report

Comments and Suggestions for Authors

Dear Authors 

The article is well revised and ready to be accepted by the journal 

Comments on the Quality of English Language

No comments 

Author Response

Thank you very much for your criticism and correction. I hope I can work with you next time.

Round 3

Reviewer 1 Report

Comments and Suggestions for Authors

The structure of the article and the link with the spices are still complicated and insufficient, so I'm keeping the decision of the 2nd review.

Comments on the Quality of English Language

.

Author Response

Thanks to the reviewer for his criticism and review. The problem that the reviewer is worried about may be the limitations of the current application of spices in liver and intestinal diseases, such as limited relevant articles and insufficient experimental results. As one of the advocates of the homology of medicine and food, I hope this article can inspire researchers to develop and utilize the potential value of spices.